# Evaluation of Biofilm Production and Antifungal Susceptibility to Fluconazole in Clinical Isolates of *Candida* spp. in Both Planktonic and Biofilm Form

**DOI:** 10.3390/microorganisms12010153

**Published:** 2024-01-12

**Authors:** Anna Marzucco, Giulia Gatti, Maria Sofia Montanari, Michela Fantini, Claudia Colosimo, Maria Vittoria Tamburini, Valentina Arfilli, Manuela Morotti, Pasqualina Schiavone, Francesco Congestrì, Martina Manera, Agnese Denicolò, Martina Brandolini, Francesca Taddei, Laura Grumiro, Silvia Zannoli, Giorgio Dirani, Alessandra Mistral De Pascali, Vittorio Sambri, Monica Cricca

**Affiliations:** 1Unit of Microbiology, The Great Romagna Hub Laboratory, 47522 Pievesestina, Italy; sofi.monta.msm@gmail.com (M.S.M.); mariavittoria.tamburini@auslromagna.it (M.V.T.); valentina.arfilli@auslromagna.it (V.A.); manuela.morotti@auslromagna.it (M.M.); pasqua.schiavone@auslromagna.it (P.S.); francesco.congestri@auslromagna.it (F.C.); martina.manera@auslromagna.it (M.M.); agnese.denicolo@auslromagna.it (A.D.); francesca.taddei@auslromagna.it (F.T.); laura.grumiro@auslromagna.it (L.G.); silvia.zannoli@auslromagna.it (S.Z.); giorgio.dirani@auslromagna.it (G.D.); vittorio.sambri@unibo.it (V.S.); monica.cricca3@unibo.it (M.C.); 2Department of Medical and Surgical Sciences—DIMEC, Alma Mater Studiorum—University of Bologna, 40126 Bologna, Italy; claudia.colosimo2@unibo.it (C.C.); martina.brandolini@outlook.it (M.B.); alessandra.depascal3@unibo.it (A.M.D.P.); 3DIN—Department of Industrial Engineering, Alma Mater Studiorum—University of Bologna, 40126 Bologna, Italy; giulia.gatti12@unibo.it; 4Health Services Research, Evaluation and Policy Unit, AUSL Romagna, 42123 Rimini, Italy; michela.fantini@auslromagna.it

**Keywords:** *Candida* spp., fluconazole, biofilm, antifungal susceptibility, blood culture

## Abstract

*Candida* spp. are an important opportunistic pathogen that can represent a possible cause of severe infections, especially in immunocompromised individuals. The clinical impact of *Candida* spp. depends, in part, on the ability to form biofilms, communities of nestled cells into the extracellular matrix. In this study, we compared the biofilm formation ability of 83 strains of *Candida* spp. isolated from blood cultures and other materials, such as respiratory samples, urine, and exudate, and their sensitivity to fluconazole (FLZ). Strains were divided into tertiles to establish cut-offs to classify isolates as low, moderate, or high biofilm producers (<0.26, 0.266–0.839, >0.839) and biofilms with low, moderate, or high metabolic activity (<0.053, 0.053–0.183, >0.183). A non-linear relationship between biofilm production and metabolic activity was found in *C. glabrata* and *C. tropicalis*. In addition, the increase in minimum biofilm eradication concentrations (MBEC_50_) compared to the Minor Inhibitory Concentration (PMIC) of the planktonic form in *Candida* spp. confirms the role of biofilm in the induction of resistance to FLZ.

## 1. Introduction

*Candida* spp. are normally commensal yeasts of the human gastrointestinal tract, mucous membranes, and skin. The breach of gastrointestinal and skin barriers, caused by factors such as trauma, infections, medical interventions, such as the insertion of an intravascular catheter, or compromised health conditions, can facilitate the onset of invasive infections, such as candidiasis [1]. The use of catheters is certainly a factor that influences the formation of biofilms, representing a greater risk in superficial and systemic fungal infections, especially in immunocompromised patients. The patient populations most at risk are the elderly, premature newborns, and those with compromised immune systems due to HIV, chemotherapy, or transplant-necessitated immunosuppression therapy [1]. Biofilms represent communities of microorganisms embedded in an extracellular polymeric substances (EPS) matrix adhered to the surface and are often associated with clinical infections [2]. 

*C. albicans* is the most frequent pathogen responsible for systemic candidiasis, followed by *C. tropicalis,* reported in Central Nervous System (CNS) infections and associated with renal microabscesses [3]. *C. parapsilosis* has emerged as the second or third leading cause of invasive candidiasis in Asia, Southern Europe, and Latin America; furthermore, it has been identified as the first or second agent of candidiasis in superficial infections [4], whereas *C. glabrata*, typically found in human skin, is mainly associated with catheter-related infections and as a major contributor to one-third of neonatal *Candida* spp. infections, with an approximate 10% mortality rate [1,5]. Despite all the advances in medical practices and diagnostic methods, hematogenous candidiasis has a crude mortality rate of approximately 50% [6].

The ability of *Candida* spp. to form biofilms is a crucial factor involved in their virulence; indeed, it prevents the penetration of substances through their dense EPS matrix, significantly reducing its susceptibility to antifungal drugs [1,7,8]. Biofilm development also weakens the host immune response by hindering macrophage phagocytosis and antibody activity [1,7], representing a severe threat to the Public Health System with serious outcomes. Typically, microbial biofilms exhibit a higher Minimum Biofilm Eradication Concentration (MBEC) compared to the normal Minimum Inhibitory Concentration of planktonic cells (PMIC) [2,9]. Increased resistance to antifungals caused by biofilms can be attributed to several factors: the expression of new regulatory genes, the presence of the EPS matrix, the presence of efflux pumps, and the existence of persister cells. These factors prevent drug absorption, induce degradation, or delay their diffusion to the innermost cell layers [9].

The specific type and dose of antifungal medication used to treat invasive candidiasis usually depends on the patient’s age, immune status, and location and severity of the infection. For most adults, the initial recommended antifungal treatment for invasive candidiasis is an echinocandin (caspofungin, micafungin, or anidulafungin) given through the vein (intravenous or IV). Fluconazole (FLZ), amphotericin B (AMB), and other antifungal medications may also be appropriate in certain situations [10].

However, the increased resistance of *Candida* spp. towards antifungals is attracting attention, especially in invasive candidiasis caused by *C. parapsilosis,* but also from *C. auris*. In this study, we considered it important to test the resistance to FLZ because it has recently been subject to an increase in resistance in *Candida* spp. [11] and to evaluate the ability of biofilm production and metabolic activity in different *Candida* spp. isolated from blood culture and materials from different body sites.

Various methods have been employed to assess biofilm formation and resistance to antifungals in clinical isolates: the air-dry method, Crystal Violet (CV) staining, and the evaluation of metabolic activity using specific reagents like Alamar Blue, (3-(4,5-dimethylthiazol-2-yl)-2,5-diphenyltetrazolium bromide) tetrazolium (MTT), or 2,3-bis-(2-methoxy-4-nitro-5-sulphenyl)-(2H)-tetrazolium-5-carboxanilide tetrazolium (XTT) [12,13]. Additionally, the presence of genes linked to biofilm formation can be investigated through molecular assays. In this view, the assessment of biofilm formation holds the possibility to improve patient therapy with the correct dose of antifungal medication, based on the susceptibility status of the infecting species.

## 2. Materials and Methods

### 2.1. Isolates

We analyzed 83 strains of *Candida* spp.: 38 *C. albicans*, 26 *C. parapsilosis*, 11 *C. glabrata,* and 8 *C. tropicalis*, isolated from various sources as blood cultures and other materials (respiratory samples, urine, and exudate) (Table 1). The specimens were collected between September 2022 and March 2023 at the O.U. Microbiology—The Greater Romagna Hub Laboratory, Cesena, Italy.

Additionally, three quality control strains of *C. albicans* (ATCC 14053™), *C. parapsilosis* (ATCC 22019™), and *C. glabrata* (ATCC MYA-2950™) were included in the study. All isolates were identified using Vitek^®^ MS (bioMérieux, Marcy l’Etoile, France).

We included in this study *Candida* spp. that are observed more frequently in episodes of invasive candidiasis, excluding *C. krusei* because it is already intrinsically resistant to FLZ.

### 2.2. Evaluation of Biofilm

#### 2.2.1. Assessment of Biofilm Production Using Air-Dry Method

The strains were initially cultured on CAN2 Chromid agar (bioMérieux, Marcy l’Etoile, France) at 37 °C for 24 and 48 h. Subsequently, the colonies of each strain were sub-cultured in Roswell Park Memorial Institute (RPMI) 1640 medium, containing L-glutamine, excluding bicarbonate (Gibco, Carlsbad, CA, USA), with a final suspension of 1 × 10^7^ CFU/mL of the sample, and were dispensed into flat-bottomed 96-well plates (SPL LifeSciences, Geumgang-ro, Pocheon, Republic of Korea). This is a fast and reliable test which allows multiple isolates to be evaluated simultaneously. 

The biomass was assessed at different times (2, 24, and 48 h) of incubation to examine the cell adhesion and the EPS matrix production. A volume of 100 µL of the sample suspension was added to each well, using the medium alone as negative control. Each strain was analyzed in triplicate. After 2 h of incubation at 37 °C, unattached cells were removed by washing three times each well using Dulbecco’s Phosphate-Buffered Saline (D-PBS) (Gibco, Carlsbad, CA, USA), followed by drying the wells to room temperature for 30 min (min). The assessment of the biomass was performed using the spectrophotometer plate reader (Digital and Analog System, Palombara Sabina, Italy) at 405 nm. 

#### 2.2.2. Assessment of Biofilm Production Using Crystal Violet Staining

The assessment of biofilm production at the 3 different times was conducted also using CV staining. Each well, after biofilm growth as descripted in Section 2.2.1, was stained with 55 μL of a 0.4% aqueous CV and incubated for 30 min. Subsequently, the wells were washed three times with 200 μL of sterile water and destained using 100 μL of 95% ethanol. After 30 min incubation, 100 μL of the destaining solution was transferred to a new plate and measured at 595 nm.

#### 2.2.3. Assessment of Metabolic Activity Using Alamar Blue Reduction Assay

The cells’ metabolic activity was assessed using an AB reagent. A volume of 100 μL of new RPMI and 10 μL of AB was added to each well after the 3 growth times (2, 24, and 48 h) and washing with D-PBS. The plates were incubated in the dark at 37 °C for 1–3 h and then read with spectrometer at 570 nm. The absorbance values of the negative control wells (without cells) were subtracted from the test values to correct any background absorbance. 

### 2.3. MBEC_50_ and PMIC Evaluation

Dilution methods, particularly the broth microdilution used in this study, allow the establishment of minimum inhibitory concentrations (MICs) of antimicrobial agents and represent the *gold standard* for antimicrobial susceptibility testing. In our study, we followed the broth microdilution protocol for fungi, according to the EUCAST model, as described by the EUCAST protocol E.Def 7.4 [14], which we will describe in the next paragraphs.

#### 2.3.1. Fluconazole Stock Solution

Scaled dilutions of FLZ (16–0.03 mg/L) (Fresenius Kabi, Bad Homburg vor der Höhe, Germany) were prepared in sterile water. These solutions were subsequently further diluted in RPMI (1X or 2X) following the EUCAST protocol E.Def 7.4 [14], utilizing the plate layout detailed in Table 2. Different types of media were implied for the biofilm and planktonic forms evaluation: RPMI-1640 1X and RPMI-1640 2X, respectively, both containing 2% glucose and 3-(N-morpholino) propane sulfonic acid (MOPS). The RPMI 2X was prepared at double strength to enable a subsequent 50% (1:1) dilution upon the addition of planktonic inoculum following EUCAST guidelines.

#### 2.3.2. Planktonic MICs (PMICs)

The susceptibility of planktonic cells to FLZ was determined using a microdilution assay following the EUCAST method [14] to establish the Planktonic Minimum Inhibitory Concentration (PMIC). A volume of 100 μL of 2X RPMI medium at a scalar concentration of FLZ with 100 μL of final planktonic suspension between 0.5 × 10^5^ and 2.5 × 10^5^ CFU/mL was inoculated into the wells. Then, they were assessed after 24 h and compared to drug-free growth controls. FLZ PMIC values were determined as the lowest drug concentration inhibiting ≥ 50% of growth compared to the drug-free control. 

The PMIC values obtained were compared with those from the microdilution assay performed using the Sensititre^TM^ YeastOne (ThermoScientific, Waltham, MA, USA) reference assay in our laboratory routine.

#### 2.3.3. Minimum Biofilm Eradication Concentrations (MBECs)

After 24 h of biofilm growth, according to the previous described protocol but without biomass assessment, the wells were washed three times with D-PBS and 100 µL of two-fold scalar diluted FLZ in 1X RPMI medium was added to the microtiter plate wells and then incubated for 24 h at 37 °C. The second-to-last and last column of the microtiter plates served as positive controls (biofilm without drug) and negative controls (medium only). After exposure to the drug, the biofilms were analyzed using a spectrophotometer and metabolic activity was determined using the AB reduction assay. MBEC_50_ for FLZ was defined as the lowest drug concentration inhibiting metabolic activity by 50%, in relation to the drug-free growth control well. 

### 2.4. Statistical Analysis

The various conditions of biofilm growth were evaluated based on the mean of three replicates and the standard deviation (SD) was determined. The results obtained were evaluated through ANOVA analysis, considering a significance level for *p*-values of 0.05. Normalization was conducted for each dataset. For each procedure, the strains were categorized into terciles according to their biofilm production and metabolic activity. This categorization established cut-off values to classify strains as having low (LBF < 0.26), moderate (MBF = 0.266–0.839), or high (HBF > 0.839) biofilm-forming capabilities and low (LMA < 0.053), moderate (MMA = 0.053–0.183), or high (HMA > 0.183) metabolic activities. All statistical analyses and graphs were performed using Stata/SE17 (StataCorp, Lakeway, TX, USA). The value of the negative control was subtracted from each absorbance of the samples.

However, in the statistical evaluation of the resistance increase from the planktonic form to the biofilm, all *Candida* strains were divided based on their susceptibility and resistance to FLZ, according to the EUCAST breakpoints [15]. By subtracting the number of strains resistant according to the PMIC from the number of strains resistant according to the MBEC_50_ and dividing by the total number of strains (resistant and susceptible), the accurate value of the resistance percentage was obtained.

## 3. Results

The results obtained were described based on the evaluation of biomass (air-dry and CV), the metabolic activity (AB), and the susceptibility to FLZ from MBEC_50_ and PMIC. The interpretation of the data was conducted taking three parameters into consideration: the species, the materials, and growth time.

### 3.1. Biofilm Quantification Assays

The results of the air-dry method performed at 2, 24, and 48 h demonstrate that the parameter with a statistically significant difference in all analyzed strains is the growth of the biomass in time (*p* = 0.0000) (Figure 1), with the only exception of *C. glabrata* with a *p* = 0.1215. All the optical densities (OD) obtained, averaging the data obtained with spectrophotometer for the three times (2, 24, and 48 h) and for each *Candida* strain, are described in Table 3. The second method, CV staining, presents the advantage of providing a more marked quantification compared to air-drying (Table 3). By this protocol, the growth time appears to be significant (*p* = 0.000) in respect to the species (*p* = 0.3897) and the material (*p* = 0.4412), while the growth analysis appears to be a significant for all species: *C. albicans p* = 0.0000, *C. parapsilosis p* = 0.0022, *C. tropicalis p* = 0.000, and *C. glabrata p* = 0.0197 (Table 3, Figure 2), unlike the air-dry method.

The distribution of isolates based on CV results was divided into terciles to establish cut-offs [16]. The *Candida* spp. with HBF both in blood cultures and other materials were *C. tropicalis* and *C. albicans*, while *C. glabrata* presents LBF in blood cultures, followed by *C. parapsilosis;* in the other material, the LBF strains were *C. parapsilosis*, followed by *C. glabrata* (Figure 3). Additional details of each strain are also shown in Appendix A.

The rank scale for biofilm producers as determined through CV staining was *C. tropicalis* > *C. albicans* > *C. parapsilosis* > *C. glabrata* for both blood cultures and other materials.

### 3.2. Biofilm Metabolic Activity by Alamar Blue Reduction Assay

The AB reduction assay was used to quantify cells of biofilm with metabolic activity, as a measure of *Candida* spp. proliferation. The distribution of values strains does not show a linear increase for any species (*p* = 0.1704) in relation to the material (*p* = 0.0604) and time (*p* = 0.0780) (Figure 4), demonstrating poor correlation of the metabolic activity with the growth time.

The *Candida* spp. with HMA from blood cultures were *C. albicans,* followed by *C. glabrata* and *C. tropicalis*, while *C. parapsilosis* presents LMA. In other materials, *C. albicans* presents HMA, followed by *C. tropicalis,* while *C. parapsilosis* and *C. glabrata* showed both 100% of LMA (Figure 5, Table 3). Additional details of each strain are also shown in Appendix A.

A poor correlation was observed between metabolic activity and the ability to form biomass. Indeed, *C. glabrata* shows an MMA and HMA relative to its LBF category and *C. tropicalis* shows an MMA relative to its HBF category. *C. albicans* and *parapsilosis*, on the other hand, show correlation.

### 3.3. Planktonic and Biofilm Susceptibility Testing

The results obtained on the sensitivity of *Candida* spp. to FLZ allow us to evaluate the MBEC_50_ and PMIC. According to the EUCAST breakpoints for yeasts [15], it was found that there was an increase in MIC > 80% for all strains of *Candida* spp. in the biofilm compared to the plaktonic form. In detail, 86.8% (33/38) in *C. albicans*, 73% (19/26) in *C. parapsilosis*, 81.8% (9/11) in *C. glabrata,* and 87.5% (7/8) in *C. tropicalis*. This increase in MIC > 80% was evaluated in all *Candida* strains that had, from the initial PMIC, an increase in MIC value of at least one serial two-fold dilution (Appendix A).

However, looking specifically at each *Candida* strain with an increase in MIC and resistance, and considering the data obtained for LBF, MBF, and HBF, we observed that the increase in biomass was often correlated with an increase in FLZ resistance, as we can see from the percentages in Figure 6. Indeed, we divided the *Candida* spp. based on their origin (blood cultures and other materials) and based on the strain, categorizing them according to their ability to form biofilms (LBF, MBF, and HBF) and their susceptibility to FLZ, according to the EUCAST breakpoints for yeasts [15]. Based on the data obtained from the statistical analysis explained in Section 2.4, we found an increase in FLZ resistance in *Candida* spp. with a greater ability to form biofilms, such as *C. tropicalis* (100%, MBF category) and *C. albicans* (75%, HBF and MBF categories), especially in blood cultures, while LBF strains (*C. parapsilosis* and *C. glabrata*) showed a slight increase in resistance (<50%). In other materials, the strains with a greater ability to form biofilms were *C. albicans* (80% and 86% in MBF and HBF categories, respectively) and *C. tropicalis* (100% MBF category). In the cases of *C. parapsilosis* (MBF category) and *C. tropicalis* (LBF category), the data did not show any increase in resistance to FLZ both in biofilm and planktonic forms (0%) (Figure 6). The rest of the data were not included because there were no strains present in that category. All the detailed results are reported in Appendix A.

## 4. Discussion and Conclusions

*Candida* spp. is recognized as a major fungal agent of nosocomial and systemic infections, where medical devices and biofilm production play a relevant role in bloodstream infections [2]. The ability of *Candide* spp. to form biofilms represents an important virulence factor that confers worse prognoses in patients [16]; therefore, its formation and the susceptibility that its presence confers to antifungals deserves a detailed study. In fact, in this study, we compared two different procedures to quantify biofilm production frequently described in the literature: the air-dry method, CV staining, and the AB reduction assay to quantify the metabolic activity of active cells. Since CV stains the metabolically active and inactive cells in mature biofilms, it is probably the most appropriate and reliable test for determining bulk biofilm formation and discriminating high, medium, and low biofilm-producing strains; in fact, in this study, we observed that the CV produces marker data, providing a clearer analysis (Table 3) [2]. 

However, by analyzing the spectrometry data (Table 3) obtained with the air-dry and CV methods, we obtained a similar trend in biomass formation in the different strains, leading to a statistically significant result on biofilm growth over time in each species. The following pattern was deduced: *C. tropicalis* > *C. albicans* > *C. parapsilosis* > *C. glabrata*, as also described by Zambrano et al. [16].

The different categories of biofilm producers were evaluated through tertile analysis, confirming the strong ability to form biofilms (MBF and HBF) by *Candida* spp., such as *C. tropicalis* and *C. albicans*, followed by a more moderate biofilm (LBF), instead, in *C. parapsilosis* and *C. glabrata*.

One of the most interesting observations when comparing the most productive biofilm strains with metabolic activity is that they seem to have no proportional correlation. 

It is not yet known which of the two characteristics (biofilm production or metabolic activity) confer on *Candide* spp. greater colonization and, consequently, complications in eradication. 

Some species associated with HBF are not associated with HMA; for example, *C. glabrata* from blood cultures showed HMA but LBF. Therefore, *C. glabrata* strains appear to be metabolically more active than cells of other species, which appear to have a high biomass; an opposite trend occurred instead in *C. glabrata* coming from other materials. This offers important view on biofilm producer strains and how the source of colony infection can influence the pathogenic potential of *Candida* strains. 

On the contrary, *C. tropicalis* is classified as HBF but MMA. In this case, the HBF could compromise the diffusion of nutrients and oxygen in the matrix due to the compact structure of the biofilm, preventing strong metabolic activity between cells, as described from scanning electron microscopy (SEM) images by Zambrano et al. [16], in which it was observed that LBF strains had less thick and compact structures, while the morphology of MBF and HBF strains was dense and covered the entire disc surface.

The *C. albicans* and *C. parapsilosis* strains instead showed good correlation between metabolic activity and biofilm production; in fact, the *C. albicans* strains are HBF but also HMA, while *C. parapsilosis* is mainly LBF with LMA.

The primary therapeutic approach for invasive infections also involves FLZ or, alternatively, polyenes and echinocandins. Nevertheless, there has been a consistent global increase in resistance in recent years, particularly in the context of nosocomial outbreaks [11]. In this study, we were able to see, as already highlighted in the literature, that *C. parapsilosis* has developed marked resistance towards FLZ over time [17]. Indeed, epidemiological data from 2018 to 2022, obtained from the analysis of bloodstream infections in Romagna, Italy, demonstrate a rise in blood infections linked to *C. parapsilosis* and an increase in azole resistance, with fluconazole resistance rising from 19% in 2018 to 52% in 2022. Regarding the susceptibility, in our study, the attention was focused mainly on the PMIC and MBEC_50_ against FLZ, evaluating a drug concentration in a range of 0.03–16 mg/L. The results show an increase in at least one serial two-fold dilution in MBEC_50_ compared to PMIC < 80% in all strains; in particular, *C. albicans* with 86.8% (33/38) and *C. glabrata* with 87.5% (Appendix A) [16]. In detail, the results of our *C. parapsilosis* isolates indicate that no significant increase in the minimum biofilm eradication concentration at 50% (MBEC_50_) was observed compared to the planktonic minimum inhibitory concentration (PMIC), considering the high initial MIC value (Appendix A), while for the other strains, an increase in biofilm resistance compared to the planktonic form was confirmed, as in the literature, and a marked correlation was found between resistance to FLZ and MBF and HBF, caused by the presence of the EPS matrix which prevents its penetration, and also due to the presence of efflux pumps and the presence of persister cells (Figure 6) [18]. This correlation is further confirmed the data observed in the literature, where an increase in resistance by the biofilm not only to FLZ but also to other antifungals, such as AMB, has been demonstrated [9,16,19].

This constitutes significant data considering that the mortality rate of patients affected by biofilm candidiasis is equal to 70% [20]. However, this correlation has been debated by several authors, such as Monfredini et al. and Atiencia-Carrera et al., reporting different mortality rates (25–70%) [6,20]. 

The results were evaluated according to the EUCAST guidelines [14]. The susceptibility values confirm an acquisition of resistance to FLZ by the biofilms in respect to the planktonic form, as can be seen from the MIC values in Appendix A.

This study represents an excellent starting point for the study of biofilms, which could be further investigated with additional antifungals and specimens from different sources. To our knowledge, there are no works in the literature comparing the growth capacity of biofilms of different *Candida* spp. coming from blood cultures and other materials, as in our case.

In conclusion, the standardization of in vitro assays has allowed us to evaluate the extent of biofilm production at different time points of *Candida* spp. from different sources and to correlate the biofilm production with the metabolic activity. It highlights the importance of assessing biofilm production with therapeutic strategies, avoiding unproductive treatments among hospitalized patients.

## Figures and Tables

**Figure 1 microorganisms-12-00153-f001:**
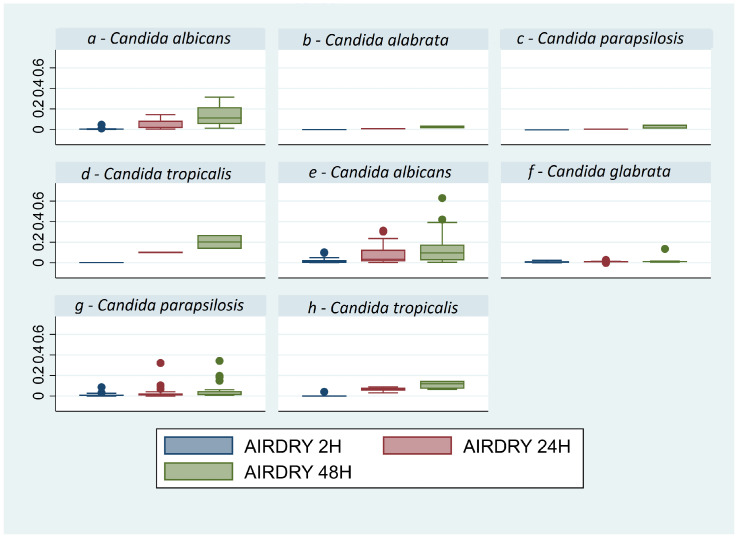
Box plot of air-dry values of *Candida* spp. for other materials (**a**–**d**) and blood cultures (**e**–**h**), the absorbance was reported on the y axis. Graph by Stata/SE17. On both materials, a constant increase over time in biomass of all strains is observed, with a small exception for *C. glabrata* (*p* = 0.1215), which shows an almost flat trend with minimal increases. The out-of-scale points in the box plot represent some isolated data obtained from strains with high capacity to form biofilm.

**Figure 2 microorganisms-12-00153-f002:**
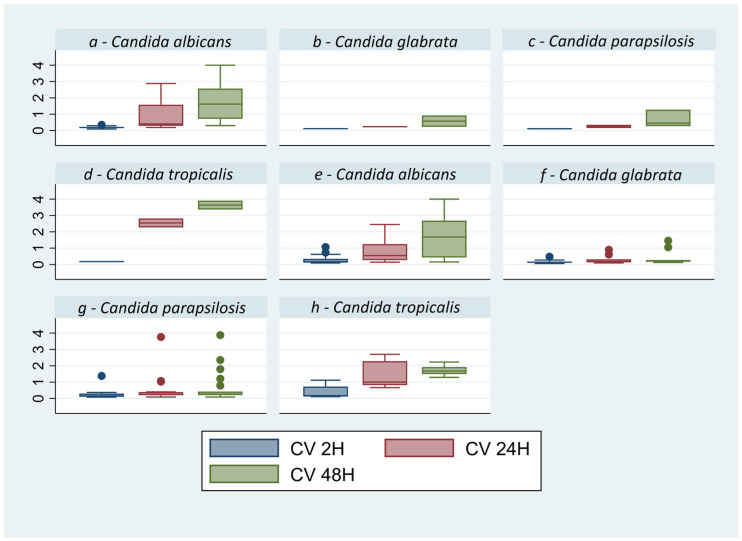
Box plot of Crystal Violet values of *Candida* spp. for other materials (**a**–**d**) and blood cultures (**e**–**h**), the absorbance was reported on the y axis. Graph by Stata/SE17. In all strains is observed a constant increase over time in biomass on both materials (*p* < 0.05). The out-of-scale points in the box plot represent some isolated data obtained from strains with high capacity to form biofilm.

**Figure 3 microorganisms-12-00153-f003:**
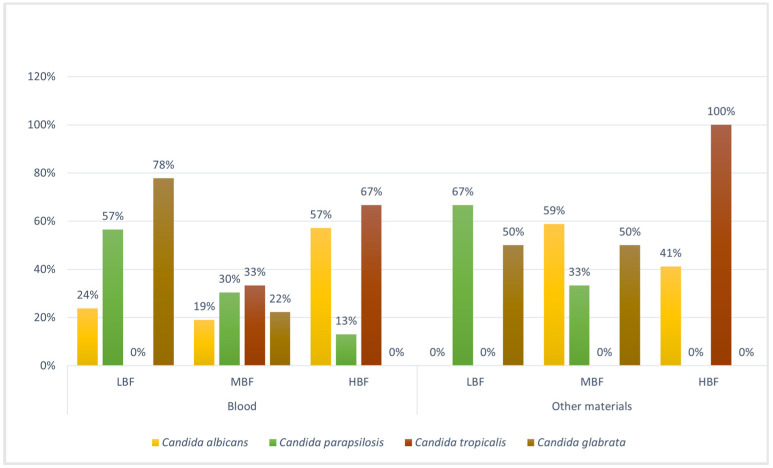
Percentage values of tertile groupings of high, medium, and low biofilm-producing strains of *Candida* spp. based on material. *C. tropicalis* and *albicans* appear to be the species with the highest number of strains in the HBF category, in contrast to *C. parapsilosis* and *glabrata*, which show high percentages of strains in the LBF category, confirming the following scale of producers: *C. tropicalis > C. albicans > C. parapsilosis > C. glabrata.* LBF: low biofilm forming; MBF: moderate biofilm forming; HBF: high biofilm forming.

**Figure 4 microorganisms-12-00153-f004:**
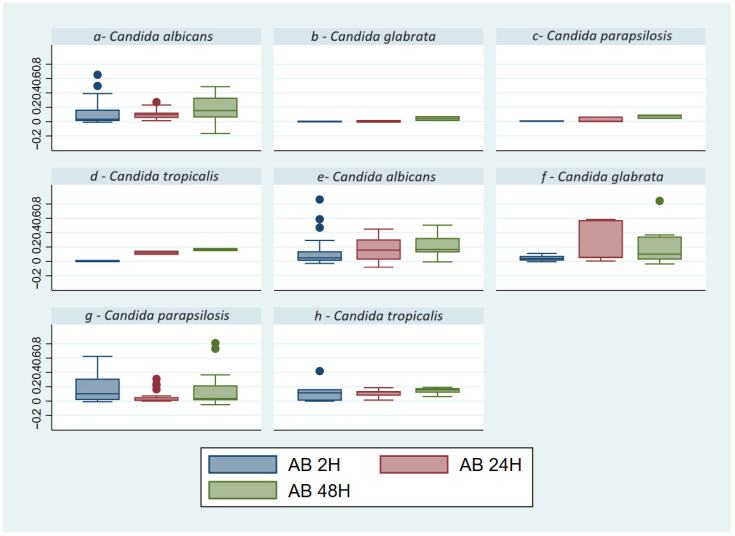
Box plot of Alamar Blue values of *Candida* spp. for other materials (**a**–**d**) and blood cultures (**e**–**h**), the absorbance was reported on the y axis. Graph by Stata/SE17. The graph shows a poor correlation of metabolic activity with the growth of Candida over time. The out-of-scale points in the box plot represent some isolated data obtained from strains with high metabolic activity.

**Figure 5 microorganisms-12-00153-f005:**
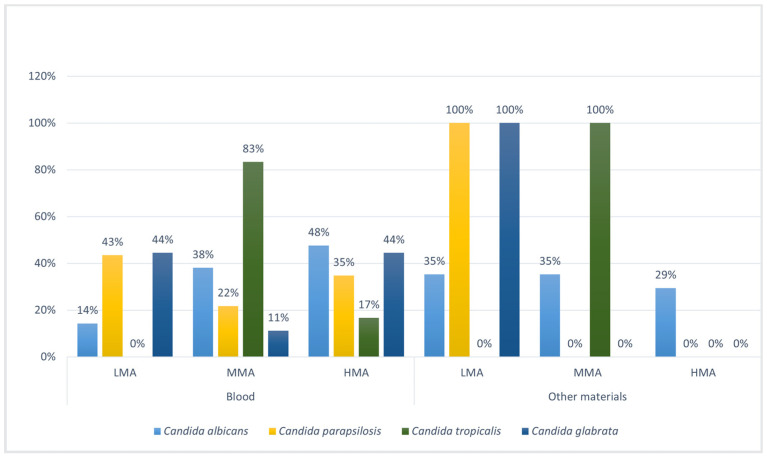
Percentage values of tertile groupings of high, medium, and low biofilm metabolic activity (HMA, MMA, LMA) of *Candida* spp. based on material.

**Figure 6 microorganisms-12-00153-f006:**
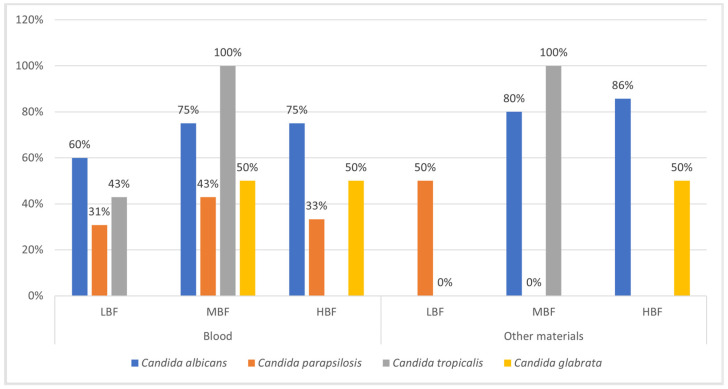
Percentage values corresponding to the increase in resistance to FLZ in the planktonic and the biofilm (LBF, MBF, HBF). The figure demonstrates a correlation between the MBF and HBF biofilm-producing strains and the increase in resistance in the biofilm compared to the planktonic form.

**Table 1 microorganisms-12-00153-t001:** Classification according to the original material of the total number of *Candida* spp. samples included in the study. Other materials: respiratory samples, urine, and exudate.

	*Candida* *albicans*	*Candida* *parapsilosis*	*Candida* *glabrata*	*Candida* *tropicalis*	Total
Blood	21	23	7	6	57
Other materials	17	3	4	2	26
Total	38	26	11	8	83

**Table 2 microorganisms-12-00153-t002:** Plate layout used in the protocol. On the x axis, the fluconazole concentrations (from 16 to 0.03 μg/mL) were reported; on the y axis, the number of sample (from 1 to 8) was reported.

	16	8	4	2	1	0.5	0.25	0.13	0.06	0.03	CP	CN
Sample 1												
Sample 2												
Sample 3												
Sample 4												
Sample 5												
Sample 6												
Sample 7												
Sample 8												

CN: negative control; CP: positive control.

**Table 3 microorganisms-12-00153-t003:** Mean absorbance values and standard deviation of the *Candida* spp. at the three times (2, 24, and 48 h) and with the three methods (air-dry, Crystal Violet and Alamar Blue).

	*Candida* *albicans* OD ± SD	*Candida* *parapsilosis* OD ± SD	*Candida* *glabrata* OD ± SD	*Candida* *tropicalis* OD ± SD
AIR-DRY 2H	0.015 ± 0.025	0.010 ± 0.018	0.007 ± 0.008	0.006 ± 0.015
AIR-DRY 24H	0.069 ± 0.084	0.030 ± 0.064	0.010 ± 0.008	0.074 ± 0.025
AIR-DRY 48H	0.137 ± 0.121	0.051 ± 0.080	0.025 ± 0.038	0.134 ± 0.065
CV 2H	0.240 ± 0.202	0.219 ± 0.253	0.163 ± 0.119	0.345 ± 0.373
CV 24H	0.846 ± 0.785	0.446 ± 0.715	0.290 ± 0.247	1.694 ± 0.910
CV 48H	1.755 ± 1.210	0.636 ± 0.861	0.456 ± 0.462	2.196 ± 0.942
AB 2H	0.142 ± 0.213	0.154 ± 0.205	0.034 ± 0.040	0.103 ± 0.143
AB 24H	0.133 ± 0.127	0.052 ± 0.084	0.182 ± 0.257	0.114 ± 0.053
AB 48H	0.195 ± 0.158	0.129 ± 0.216	0.184 ± 0.260	0.150 ± 0.044

OD ± SD: Optical density ± Standard Deviation; CV: Crystal Violet; AB: Alamar Blue; H = hours.

## Data Availability

The data presented in this study are available on request from the corresponding author. The data are not publicly available due to privacy.

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
