# Peer review of "Evaluation of Biofilm Production and Antifungal Susceptibility to Fluconazole in Clinical Isolates of Candida spp. in Both Planktonic and Biofilm Form"

_microorganisms, 2024, doi:10.3390/microorganisms12010153_

Round 1

Reviewer 1 Report

Comments and Suggestions for Authors

Dear authors,here are my questions:

1.According to what did you choose the tested concentrations of fluconazole?

2. Why did you choose and test only fluconazole? There is plenty of antimycotics,you mentioned som of them in the introduction.

3. In discussion,there is lack of references,which compare your results with previous findings. 

Discssion should more compare.

4. Did you find some important differences between your data and previous data,like changes in PIMC,MBEC? These changes can show us possible mutations in strains to addapt on host organisms.

Author Response

Dear Reviewer 1,

Reviewer 2 Report

Comments and Suggestions for Authors

In this study, the authors evaluate the biofilm formation ability of 83 strains of Candida and their sensitivity to fluconazole.

The overall writing of this manuscript is flawed, and some results are presented repeat in the Table and Fig. In addition, Some important information are missing, such as the MICs of each strains and how to determine the increase of resistance. Whats the objectives of this study? Some sentences are very difficult to understand.

Minor issues:

The quality of Figs are poor, and the results of Statistical analysis are not presented in Figs or Tables; ;

Line 4, forms. to form

Line 76-87, suggested the two paragraphs merged into one paragraph;

Line 120, read-er to reader;

Line 122, also with using, delete with;

Author Response

Dear Reviewer 2,

Round 2

Reviewer 2 Report

Comments and Suggestions for Authors

I have no more comments on this revised manuscript.